

# Hydroxycinnamoyl-coenzyme A: tetrahydroxyhexanedioate hydroxycinnamoyl transferase (HHHT) from *Phaseolus vulgaris* L.: phylogeny, expression pattern, kinetic parameters, and active site analysis

Amanda Fanelli[1,2], Christina Stonoha-Arther[1] and Michael L. Sullivan[1]

[1] Agricultural Research Service, US Dairy Forage Research Center, United States Department of Agriculture, Madison, Wisconsin, United States
[2] Oak Ridge Institute for Science and Education, Oak Ridge, Tennessee, United States

Corresponding author
Amanda Fanelli,
amanda.fanelli@usda.gov

## ABSTRACT

BAHD acyl-coenzyme A (CoA) transferases comprise a large family of enzymes in plants which transfer an acyl group from a CoA thioester to hydroxyl or amine groups to form esters or amides, respectively. Clade Vb of this family primarily utilizes hydroxycinnamoyl-CoA as the acyl donor. These enzymes are involved in biosynthesis of diverse specialized metabolites with functions such as structure (*e.g.*, lignin formation) and biotic/abiotic stress mitigation. The diversity of these enzymes has arisen from both divergent and convergent evolution, making it difficult to predict substrate specificity or enzyme function based on homology, and relatively few BAHD transferases have been characterized biochemically with respect to substrate specificity. We previously identified a hydroxycinnamoyl-CoA: tetrahydroxyhexanedioate hydroxycinnamoyl transferase (HHHT) from common bean capable of transferring hydroxycinnamic acids to mucic or saccharic acid to form the corresponding esters. Here, to better understand the structure/function relationships of this enzyme, we have further characterized it with respect to expression pattern, kinetic parameters, and predicted three-dimensional (3-D) structure and active site interactions with acceptor substrates. The *hhht* gene was expressed predominantly in leaves and to a lesser extent flowers and shoots. $K_M$ values did not vary greatly among donor or among acceptor substrates (generally less than two-fold), while $k_{cat}$ values were consistently higher for saccharic acid as substrate compared to mucic acid, leading to higher catalytic efficiency (as $k_{cat}/K_M$) for saccharic acid. Both acceptors had similar binding poses when docked into the active site, and the proximity of multiple hydroxyl groups to the catalytic His 150, especially for saccharic acid, might provide some insights into regiospecificity. These findings provide a foundation for better understanding how the 3-D structure of BAHD transferases relates to their substrate specificity, as we explore the chemistry of the active site and interactions with ligands. This could ultimately lead to better prediction of their function and ability to rationally design BAHD transferases to make useful and novel products.

## INTRODUCTION

Acylation is an important step in the synthesis of several plant metabolites. Many of these processes involve enzymes belonging to the BAHD acyltransferase family, which catalyze the transfer of an acyl group from a coenzyme A (CoA) thioester to alcohol or amine groups, producing esters or amides, respectively (*Bontpart et al., 2015*). BAHD acyltransferases participate in the synthesis of anthocyanins, flavonoids, cell wall components, and other specialized metabolites (*Moghe et al., 2023*). Phylogenetic analyses have divided this family into eight clades (*Tuominen, Johnson & Tsai, 2011*) named Ia, Ib, II, IIIa, IIIb, IV, Va, and Vb. There is evidence of both divergent and convergent evolution in the BAHD family. Consequently, similarity of amino acid sequence does not always indicate similar substrate specificities. There are enzymes that share high sequence similarity but use different substrates (*Kruse et al., 2022*; *Sullivan & Knollenberg, 2021*), and also enzymes with low sequence similarity that have similar catalytic roles (*Fu et al., 2022*). Therefore, it is difficult to use homology to predict function (*Luo et al., 2007*). Furthermore, a limited number of BAHD enzymes have had their three-dimensional (3-D) structure elucidated or were characterized with respect to kinetic parameters and substrate specificity, which also contributes to the difficulty of predicting the catalytic function based on the amino acid sequence. Nevertheless, the recent improvement of algorithms for 3-D protein structure prediction and molecular docking is promising to allow a better understanding of the relationship between structure and substrate specificity in the BAHD family (*Jisna & Jayaraj, 2021*; *Fanelli & Sullivan, 2022*).

In clade Vb of BAHD acyltransferases, characterized enzymes catalyze reactions in which aromatic acids, predominately hydroxycinnamates, are the donated acyl group. The Vb enzymes have roles in the synthesis of compounds related to plant development and defense, such as lignin precursors, phaselic acid, hydroxycinnamoyl spermidines, and phytoalexins. Therefore, this clade is an attractive target for many biotechnological and metabolic engineering applications (*Yang et al., 2004*; *Grienenberger et al., 2009*; *Hoffmann et al., 2005*; *Sullivan, 2009*).

In red clover (*Trifolium pratense* L.), we have shown that two clade Vb enzymes, HMT and HDT, are responsible for the accumulation of the caffeoyl derivatives phaselic acid (2-*O*-caffeoyl-L-malic acid) and clovamide (*N*-caffeoyl-L-3,4-dihydroxyphenylalanine) (*Sullivan & Zarnowski, 2011*; *Sullivan & Knollenberg, 2021*; *Sullivan, 2009*). The oxidation of these compounds by polyphenol oxidases (PPO) may have a role in the defense against insect herbivores and pathogens (*Constabel & Barbehenn, 2008*). This process also has applications in improving forages for animal feed, as the oxidation of these *o*-diphenols by PPO reduces post-harvest protein degradation after ensiling (*Sullivan & Hatfield, 2006*). The PPO/*o*-diphenol system could be transferred to other forages, such as alfalfa, that do not naturally possess it to mitigate protein losses when ensiled. However, we have demonstrated that red clover HMT has a greater than five-fold preference ($V_{max}/K_M$) for
*p*-coumaroyl- or feruloyl-CoA as an acyl donor over caffeoyl-CoA (*Sullivan & Zarnowski, 2011*), and overexpression of red clover HMT in alfalfa (*Medicago sativa* L.) results in the accumulation of mostly *p*-coumaroyl- and feruloyl-malate, not phaselic acid (*Sullivan, Green & Verdonk, 2021*). These *p*-coumaroyl and feruloyl derivatives are ineffective at preventing protein degradation (*Sullivan & Zeller, 2013*). In the search for an HMT from *Phaseoulus vulgaris* L. (common bean, hereafter referred to as bean) that might have a stronger preference for caffeoyl-CoA, we identified a hydroxycinnamoyl-CoA: tetrahydroxyhexanedioic acid hydroxycinnamoyl transferase (HHHT), which transfers hydroxycinnamates to mucic and saccharic acid acceptors to form the corresponding esters (Fig. 1) (*Sullivan, 2017*).

Although it has been suggested that these compounds may also be involved in plant defense, little is known about the role of HHHT in the bean plant. Furthermore, the kinetic parameters for this enzyme have not been measured, and it is unknown whether HHHT prefers a specific acyl donor or acceptor. Along with kinetic data, elucidating the enzyme's 3-D structure and active site may give information about how enzyme structure influences substrate preference, which could be used for the rational design of BAHD enzymes. In this study, we sought to investigate the evolutionary relationship of bean HHHT with other enzymes in clade Vb and gain insights about its function, enzyme kinetics, structure, and active site.

## MATERIALS AND METHODS

### Annotation of BAHD transferases and phylogenetic analysis

To annotate putative BAHD acyltransferases in *Phaseolus vulgaris* L., *Trifolium pratense* L., *Medicago truncatula* Gaetrn., *Arabidopsis thaliana* (L.) Heynh., *Brachypodium distachion* (L.) P.Beauv. and *Panicum Hallii* Vasey, the hmmsearch algorithm was used on the EMBL-EBI website (https://www.ebi.ac.uk/Tools/hmmer/search/hmmsearch) (*Potter et al., 2018*). An accession search was performed using PF02458 at PFAM (transferase) profile against the Ensembl proteomes database, restricting to the species mentioned. From all sequences returned, only those with an e-value $< 10^{-5}$ were considered as true positive results. Isoforms (that had duplicate names in the database) were identified using a Python script, and only one sequence per coding gene was kept for analysis. To annotate clade Vb from BAHD acyltransferases, a phylogenetic analysis was performed with the set of transferases identified by hmmsearch and a set of 69 characterized BAHD proteins (*Tuominen, Johnson & Tsai, 2011*), which were downloaded from NCBI. The two sets were compared to remove any duplicates. These sequences were aligned using MUSCLE5 in seaview5 software (*Gouy et al., 2021*). The alignment was inspected for the presence of the active site HXXXD motif. Sequences that had this region conserved were kept for analysis. We also included sequences that had a residue other than H in the first position of the motif, as a biochemically characterized BAHD has this His replaced by Ser (*Walker, Long & Croteau, 2002*). To place the root in the phylogenetic tree, a set of three transferase sequences from fungi (RHIMIDRAFT_244343 from *Rhizopus microspores*, DM01DRAFT_1381931 from *Hesseltinella vesiculosa*, and BCR42DRAFT_237763 from *Absidia repens*) were annotated using hmmsearch as above and included in the analysis as

**Figure 1 Donor and acceptor substrates for HHHT.** HHHT is capable of transferring a *trans*-hydroxycinnamic acid (here *p*-coumaric: $R_1$ = H, $R_2$ = OH; caffeic: $R_1$ = OH, $R_2$ = OH; ferulic $R_1$ = $OCH_3$, $R_2$ = OH) from coenzyme A (CoA) to a hydroxyl group on mucic or saccharic acid to form an ester.

an outgroup. After obtaining this final set, all sequences were aligned using MUSCLE5 in seaview5 software. The alignment was manually edited, and a maximum likelihood phylogeny was built using the IQ-Tree webserver (http://iqtree.cibiv.univie.ac.at/) (*Trifinopoulos et al., 2016*). Branch support was obtained using SH-aLRT. The best-fit model, chosen by ModelFinder (*Kalyaanamoorthy et al., 2017*) using BIC criteria, was VT + F + G4. The tree was visualized using iTOL (*Letunic & Bork, 2021*). After identifying clade Vb in the BAHD phylogeny, another phylogenetic tree was inferred only for the members of this clade, with four sequences from other clades used as an outgroup to root the tree, plus a set recently characterized (Table S1). The parameters were as before, and the best-fit model chosen was JTT + I + G4. The final tree was visualized with iTOL and edited using Figma.

## Gene expression and *cis*-element promoter analysis

We analyzed RNA-seq data from *O'Rourke et al. (2014)* available at https://www.zhaolab.org/PvGEA/, and from *Ayyappan et al. (2015)*. The work from *O'Rourke et al. (2014)* provided RNA-seq data for the *Phaseolus vulgaris* cultivar 'Negro Jamapa'. Samples consisted of seven tissues at specific developmental stages, and three nitrogen treatments that consisted of inoculating nodules with the effective nitrogen fixing strain *Rhizobium tropici* CIAT899, or with the ineffective strain *Rhizobium giardini* 6917, or providing fertilizer containing adequate levels of $NO_3^-$ for growth. The work from *Ayyappan et al. (2015)* analyzed RNA-seq data for leaves of the bean cultivar 'Sierra' inoculated with the fungal rust *Uromyces appendiculatus*. Time points for sample collection were 0, 12, and 84 h after inoculation. The meta-analysis and visualization in our work here were performed using R. The search for *cis*-elements in the promoter region of the *Pvhhht* gene was conducted by using the 600 bp region upstream as a query in the newPLACE database (https://www.dna.affrc.go.jp/PLACE/?action=newplace) (*Higo et al., 1999*).

## Expression of *Phaseolus vulgaris* HHHT in *Escherichia coli* and purification of the His-tagged protein

DNA encoding HHHT was synthesized by GenScript (Piscataway, NJ, USA) based on the predicted amino acid sequence (GenBank AOX15526.1) of the previously characterized

cDNA (GenBank KX443573) with codon optimization for *E. coli* using the supplier's algorithm and specifying *Nde*I and *Xho*I be absent from the open reading frame. The sequence CAT was added to the 5′ end to create an *Nde*I site, and a TAA stop codon and *Xho*I site were added to the 3′ end. The synthesized DNA (GenBank ON240067) was cloned as an *Nde*I-*Xho*I fragment between those restriction sites of pET28a(+) (MilliporeSigma, Burlington, MA, USA). The resulting construct fuses a 6xHis-tag to the N-terminus of HHHT.

The pET28-based construct was transformed into Rosetta 2 (DE3) pLysS competent cells (MilliporeSigma). Transformed cells were grown and induced, and a lysate of the cells was made essentially as described in the pET System Manual (available at http://emdmillipore.com as TB055). A 1 L culture in LB medium with 50 μg/mL of kanamycin and 34 μg/mL chloramphenicol was grown with shaking at 37 °C to an $OD_{600nm}$ of 0.54, induced with isopropyl ß-D-1-thiogalactopyranoside (IPTG) added to 1 mM, then grown an additional 20 h at 18 °C. The cells were harvested by centrifugation, resuspended in 20 mL Bugbuster reagent (MilliporeSigma) to which 10 μL benzonase nuclease and 200 μL protease inhibitor cocktail were added (70746 and P8849, respectively; MilliporeSigma), and incubated at room temperature for 20 min. Insoluble material was removed by centrifugation at 17,000× $g$ for 10 min at 4 °C. To the 20 mL of cleared lysate, 80 mL of 1.25 X binding buffer (1 X is 20 mM sodium phosphate, pH 7.5, 500 mM NaCl, 20 mM imidazole) was added. The non-proteinaceous precipitate that formed upon binding buffer addition was removed by centrifugation at 30,000× $g$ for 10 min at 4 °C, and the clarified sample was added to 1 mL Ni-Sepharose 6 Fast Flow (Catalog 17-5318-01; GE Healthcare BioScience AB, Uppsala, Sweden) that had been washed and pre-equilibrated with binding buffer. Tagged protein was allowed to bind to the Ni-Sepharose by incubating at room temperature on a nutator for 80 min. The Ni-Sepahrose was transferred to a small column, washed five times with 1 mL of binding buffer, and the protein was eluted with 3 mL of 250 mM imidazole in 20 mM sodium phosphate, pH 7.5, 500 mM NaCl. The eluate was dialyzed overnight against 500 mL 20 mM sodium phosphate, pH 7.5, 500 mM NaCl at 4 °C. The sample was further concentrated, and the buffer was exchanged to 100 mM sodium phosphate, pH 7.5 using an Amicon Ultra-15 concentrator (MilliporeSigma) with a 10,000 molecular weight cutoff. Aliquots were made, flash-frozen in liquid nitrogen, and stored at −60 °C until needed for analyses. The concentration of the purified protein was estimated by comparison to known amounts of bovine serum albumin on SDS-PAGE gels.

In addition to the wild-type HHHT, four putative active site mutant gene constructs were synthesized (T35A, H150A, W386A, and R414A) by replacing the corresponding codons of the codon-optimized version of the gene with GCG. These were transformed and expressed in *E. coli*, and protein was purified as described above for the wild type.

### Determination of HHHT kinetic parameters

Kinetic parameters for HHHT were determined by measuring initial reaction rates by release of free CoA in near-real time using DTNB (5,5′-dithio-*bis*- (2-nitrobenzoic acid)) and spectroscopy as detailed elsewhere (*Sullivan & Bonawitz, 2018*; *Sullivan, 2023*). Measurements were made in a temperature-controlled spectrophotometer at 25 °C.

Reactions (1 mL final volume) were carried out in 100 mM sodium phosphate, 1 mM EDTA, 0.2 mM DTNB. Acceptor substrates were prepared as 50 mM stock solutions of the dipotassium salt by adding two equivalents of KOH to mucic acid (M89617; MilliporeSigma) and one equivalent of KOH to the saccharic acid monopotassium salt (S4140; MilliporeSigma), to enhance solubility. The substrate and enzyme amounts used are detailed in Table 1.

All kinetic data analyses were carried out using GraphPad Prism version 8 for Mac OS X (GraphPad Software, La Jolla, CA, USA). Each analysis (*i.e.*, series of substrate concentrations) was carried out in duplicate with freshly prepared substrate and enzyme dilutions. Data were analyzed by non-linear regression of the replicated data. All donor/acceptor substrate combinations were fit with the Michaelis-Menten enzyme kinetics model. Kinetic parameters are reported ± standard error (SE) as determined from the non-linear regression.

### Structure prediction and molecular docking

A *pdb* file (AF-V7BKA6-F1-v4) with the coordinates of the predicted three-dimensional structure of PvHHHT was downloaded from the AlphaFold website (https://alphafold.ebi.ac.uk/). The quality of the model was evaluated using MolProbity (*Williams et al., 2018*). The molecules of the two acyl acceptor ligands, saccharic acid and mucic acid, were rendered using Avogadro (v 1.2.0). The protonation state used was for a pH of 7.4. A conformer search was performed to find the minimum energy conformation, as described (*Fanelli & Sullivan, 2022*). The final coordinates were saved in *mol2* files. AutoDockVina (v 1.2.5) (*Eberhardt et al., 2021*) was used to perform the molecular docking of the ligands into the enzyme. To generate the *pdbqt* files required by Vina, the Python Meeko package was used for the ligands, using the mk_prepare_ligand.py script. For the protein, the ADFR software suite (*Ravindranath et al., 2015*) was used. For the docking, the coordinates of the center were set to be the center of mass of the HXXXD active site motif, as described (*Fanelli & Sullivan, 2022*), the search space was 20 × 20 × 20 Å, and the exhaustiveness was set to 32, therefore allowing the bonds to rotate. The results were visualized using Pymol. To perform docking for the acyl donors (*p*-coumaroyl, caffeoyl-, and feruloyl-CoA), we used only a portion of the molecule (N-acetyl-S-hydroxycinnamoyl-cysteamine), since the CoA is too large to be handled by the Vina algorithm. The molecules were rendered and the docking was performed as described for the acceptors.

## RESULTS AND DISCUSSION

### Phylogenetic analysis of clade Vb of BAHD acyltransferases

To explore the evolution of BAHD acyltransferases in bean, particularly clade Vb, relative to other species, we identified BAHD sequences in the proteomes of four dicots (bean, red clover, *Medicago truncatula*, Arabidopsis) and two grasses (*Panicum hallii* and *Brachypodium distachion*), and analyzed their phylogenetic relationship to a set of biochemically characterized BAHD enzymes (Table 2). The genomes analyzed here encode a large number of BAHD acyltransferases, consistent with this family having several

**Table 1  Reaction conditions for determination of kinetic parameters.**

| Variable substrate | Range (μM) | Number of conditions | Constant substrate | HHHT protein (μg/mL) |
|---|---|---|---|---|
| p-Coumaroyl-CoA | 1 to 100 | 10 | Mucic acid (20 mM) | 2.0 |
| p-Coumaroyl-CoA | 1 to 100 | 10 | Saccharic acid (20 mM) | 0.6 |
| Caffeoyl-CoA | 1 to 100 | 10 | Mucic acid (20 mM) | 3.0 |
| Caffeoyl-CoA | 1 to 100 | 11 | Saccharic acid (20 mM) | 1.0 |
| Feruloyl-CoA | 1 to 100 | 10 | Mucic acid (20 mM) | 1.5 |
| Feruloyl-CoA | 1 to 100 | 11 | Saccharic acid (20 mM) | 0.8 |
| Mucic acid | 500 to 20,000 | 9 | p-Coumaroyl-CoA (100 μM) | 2.0 |
| Mucic acid | 500 to 20,000 | 10 | Caffeoyl-CoA (100 μM) | 3.0 |
| Mucic acid | 500 to 20,000 | 10 | Feruloyl-CoA (100 μM) | 1.5 |
| Saccharic acid | 250 to 20,000 | 11 | p-Coumaroyl-CoA (100 μM) | 0.6 |
| Saccharic acid | 250 to 20,000 | 11 | Caffeoyl-CoA (100 μM) | 1.0 |
| Saccharic acid | 250 to 20,000 | 11 | Feruloyl-CoA (100 μM) | 0.8 |

**Table 2  Total BAHD acyltransferases and number per clade identified in the proteomes of indicated species.**

| Species | Total | Ia | Ib | II | IIIa | IIIb | IV | Va | Vb |
|---|---|---|---|---|---|---|---|---|---|
| *Panicum hallii* | 107 | 15 | 10 | 4 | 0 | 9 | 20 | 38 | 11 |
| *Brachipodium diystachion* | 93 | 15 | 10 | 4 | 0 | 8 | 13 | 30 | 13 |
| *Medicago truncatula* | 133 | 28 | 35 | 5 | 11 | 6 | 0 | 34 | 14 |
| *Trifolium pratense* | 84 | 18 | 10 | 6 | 10 | 3 | 0 | 24 | 13 |
| *Arabidopsis thaliana* | 62 | 11 | 14 | 5 | 9 | 4 | 0 | 16 | 3 |
| *Phaseolus vulgaris* | 78 | 13 | 14 | 4 | 7 | 3 | 0 | 21 | 16 |

distinct roles in plants, such as the synthesis of anthocyanins/flavonoids (clade Ia), epicuticular waxes (clade II), volatile esters (clade Va), acylation of oligosaccharide sugars, alkaloids, and terpenes (clade IIIa), aliphatic amine acylation (clade IV), among other functions (*Moghe et al., 2023*). As observed in other studies, the distribution among clades varies depending on the species, and clade IV seems to be predominantly present in grasses, whereas clade IIIa seems specific to dicots (*Tuominen, Johnson & Tsai, 2011*; *Bartley et al., 2013*).

We then performed a more detailed analysis of clade Vb (Fig. 2). In this clade, we identified two major subclades. One contains mostly HCT/HQT sequences, which are involved in the synthesis of monolignols and chlorogenic acid, but also includes AsHHT from oats (*Yang et al., 2004*) and DcHCBT from *Dianthus caryophyllus* (*Yang et al., 1997*), which transfer aromatic groups, such as benzoyl and hydroxycinnamoyl, to anthranilates, producing amides that are precursors of phytoalexins. It also contains the recently identified EpHMT from *Echinacea purpurea*, which transfers hydroxycinnamates to malic acid, producing phaselic acid. The other subclade is expanded in dicots, with no representative of the grasses *Panicum halli* and *Brachypodium distachion*, which suggests
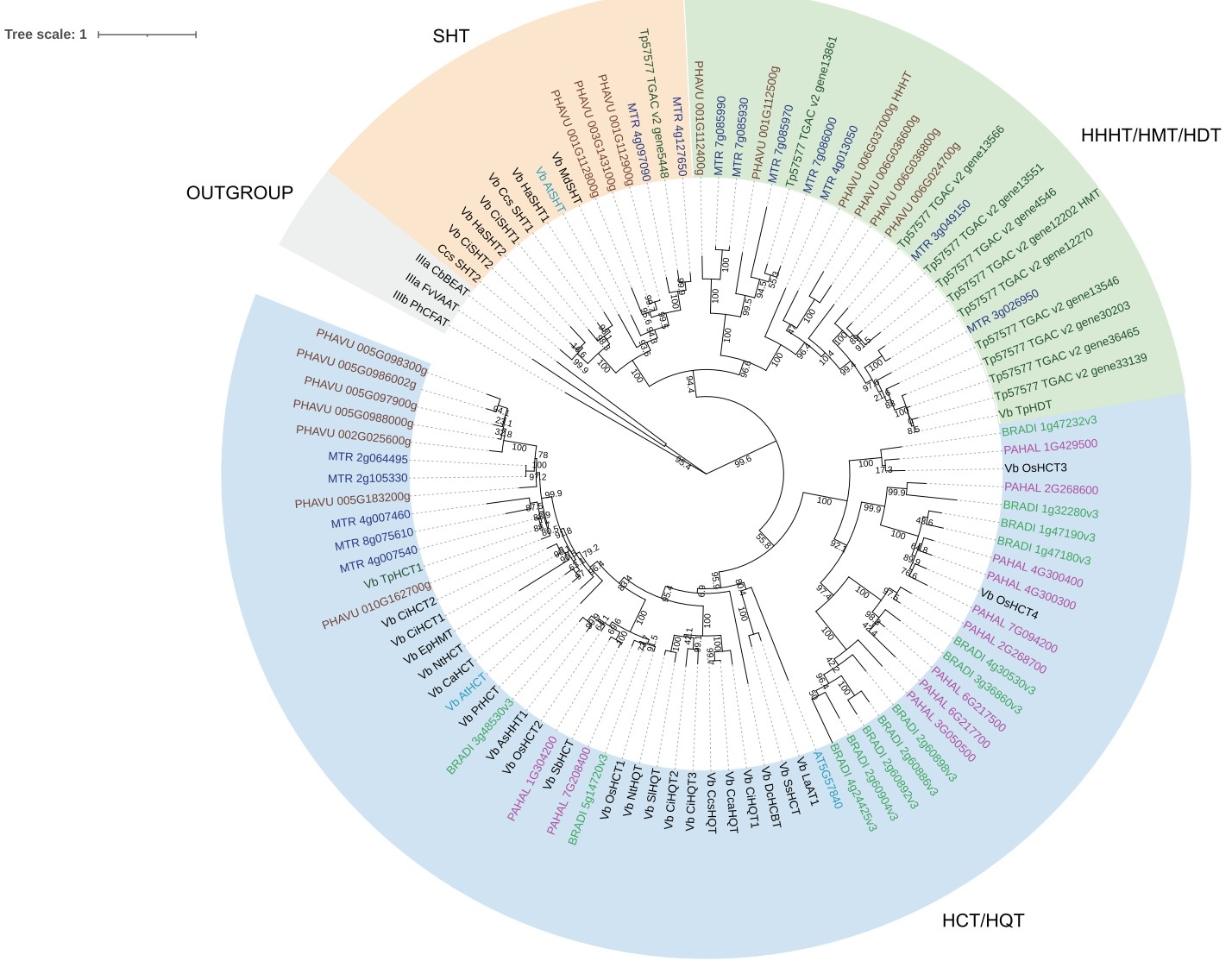

**Figure 2 Maximum likelihood phylogeny of clade Vb of BAHD acyltransferases.** Sequences in brown are from *Phaseolus vulgaris*, dark blue from *Medicago truncatula*, dark green from *Trifolium pratense*, light blue from *Arabidopsis thaliana*, green from *Brachypodium distachion* and pink from *Panicum hallii*.

that these enzymes may have roles specific to dicots. This subclade further subdivides into two. One contains spermidine hydroxycinnamoyl transferases (SHT) identified in dicots such as *Arabidopsis thaliana*, *Helianthus annus* and *Malus domestica*. These characterized enzymes are involved in the synthesis of amides in flowers (*Grienenberger et al., 2009*; *Elejalde-Palmett et al., 2015*; *Li et al., 2021*). We identified sequences from bean, red clover, and *Medicago truncatula* belonging to this subclade, which could have similar function. However, considering the occurrence of both convergent and divergent evolution in the BAHD acyltransferases, it is difficult to predict their role. The other subclade contains the red clover TpHMT and TpHDT (*Sullivan, 2009*; *Sullivan & Knollenberg, 2021*), and the bean PvHHHT. Although these three enzymes belong to the same subclade, they form

different products. TpHMT utilizes malic acid as the hydroxycinnamoyl acceptor and it is essential for the accumulation of phaselic acid (caffeoyl-malate) and other hydroxycinnamoyl-malate esters in red clover (*Sullivan & Zarnowski, 2011*). TpHDT utilizes L-tyrosine and L-DOPA (L-3,4-dihydroxyphenylalanine) as hydroxycinnamoyl acceptors and has a role in the synthesis of clovamide and related amides (*Sullivan & Knollenberg, 2021*). The role of phaselic acid and clovamide in plants is not fully understood, but they are oxidized by polyphenol oxidases (PPO), generating products that may be involved in the defense against insect herbivores and pathogens (*Thipyapong, Hunt & Steffens, 2004*). PvHHHT utilizes tetrahydroxyhexanedioic acid acceptors mucic and saccharic acid, forming hydroxycinnamoyl esters with them (*Sullivan, 2017*). The role of these compounds has not yet been elucidated, but they also may be involved in biotic/abiotic stress responses (*Elliger et al., 1981*).

Comparing the amino acid sequences, PvHHHT shares 70% similarity (54% identity) with TpHMT and 71% similarity (56% identity) with TpHDT while TpHMT and TpHDT share 83% similarity (72% identity) (Fig. 3). Even though the three enzymes are similar with respect to amino acid sequence, they utilize different acyl acceptor substrates and make distinct products. This functional divergence is common in the BAHD acyltransferase family (*Kruse et al., 2022*). On the other hand, there are also examples of BAHD enzymes with little sequence similarity that have the same substrate specificity. Two ferulate monolignol transferases, one from *Angelica sinensis* Oliv. and another from rice share only 20% similarity. The enzyme from the eudicot *A. sinensis* Oliv. belongs to clade IIIa, whereas the one from rice belongs to clade Va (*Karlen et al., 2016*). Also, the EpHMT enzyme from *Echinacea purpurea* belongs to the HCT/HQT subclade of clade Vb (*Fu et al., 2022*), sharing only 55% similarity (36% identity) with the red clover HMT.

Although we have measured HMT activity in leaves of bean (*Sullivan, 2017*), an HMT gene has yet to be identified. The other bean proteins in the HMT/HDT/HHHT subclade are potential candidates. However, considering the many cases of functional divergence and convergent evolution in the BAHD family, the bean HMT enzyme could also belong to another clade of BAHD acyltransferases.

## Expression profiling and promoter analysis

To investigate the expression pattern of *Pvhhht* in distinct tissues and conditions, we retrieved data from the *P. vulgaris* Gene Expression Atlas (*O'Rourke et al., 2014*), which provides gene expression levels obtained from seven tissues at distinct developmental stages and under three nitrogen treatments. The *Pvhhht* gene was highly expressed in leaves and flowers, a finding consistent with isolation of the cDNA from young leaves (*Sullivan, 2017*). Furthermore, the highest levels of expression were in leaves 21 days after inoculation with an ineffective nitrogen-fixing strain of rhizobia, which suggests an upregulation of *Pvhhht* under the condition of nitrogen deficiency (Fig. 4A). We also analyzed data from another work (*Ayyappan et al., 2015*) that identified differentially expressed genes in bean leaves inoculated with the fungal rust *Uromyces appendiculatus*. In this dataset, *Pvhhht* was upregulated 84 h after inoculation (Fig. 4B). Taken together with

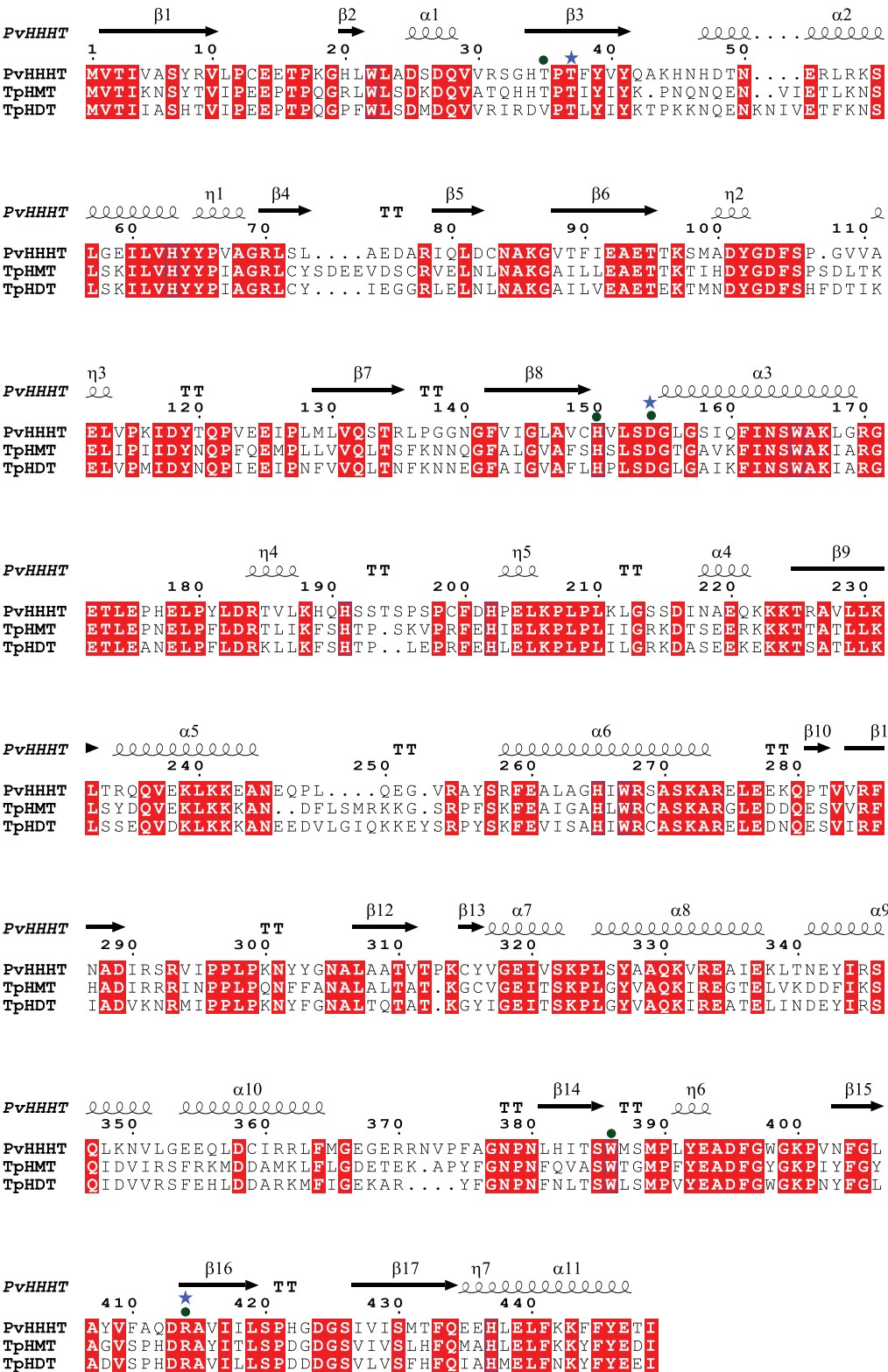

**Figure 3 Protein sequence alignment showing the secondary structure of *Phaseoulus vulgaris* HHHT, as predicted by AlphaFold, and the sequence similarity with *Trifolium pratense* HMT and HDT.** Residues highlighted in red are conserved in all three proteins. Alignment was performed by MUSCLE and visualized using ESPipt3. Residues in the active site interacting with acyl donor substrates are marked with a circle, and those interacting with acyl acceptors are marked with a star.

A

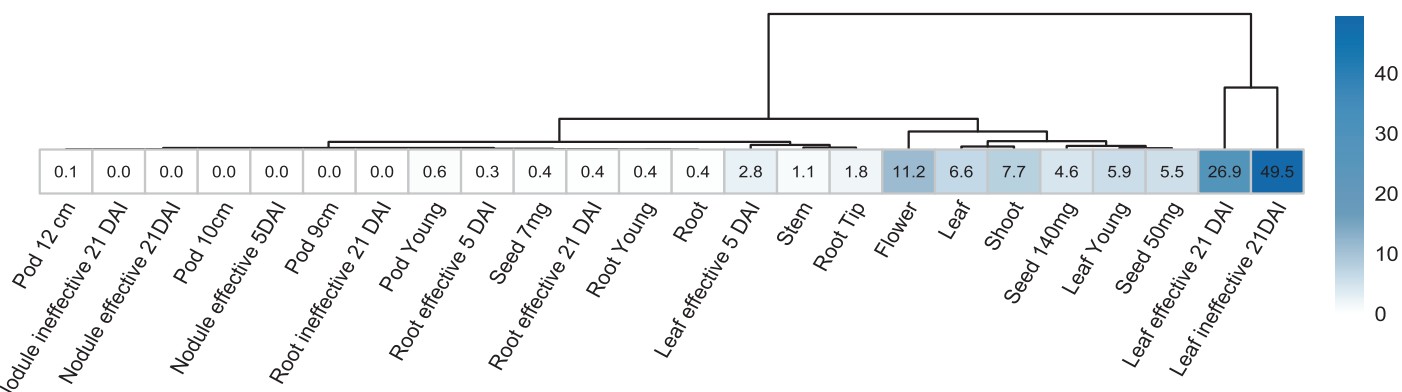

B

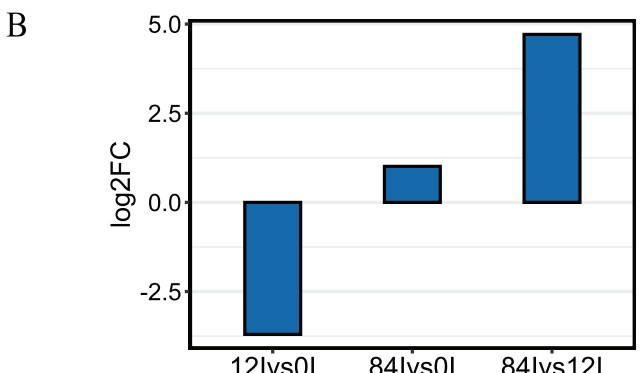

**Figure 4 Expression of *Pvhhht*.** (A) Expression profile of *Pvhhht* in pods, roots, leaves, stems, and seeds of *P. vulgaris* cultivar 'Negro Jamapa' subjected to regular fertilization, or inoculated with effective nitrogen fixator strain *Rhizobium tropici* CIAT899, or with the ineffective *Rhizobium Giardini* 6917. Values are mean FPKM obtained using RNA-seq. Data are from *O'Rourke et al. (2014)*. (B) Expression profile of *Pvhhht* in leaves of *P. vulgaris* cultivar 'Sierra' after inoculation with the rust pathogen *U. appendiculatus*. Expression values were obtained using RNA-seq and are the log2 fold change at 12 *vs.* 0 h post-inoculation (12Ivs0I), 84 *vs.* 0 h post inoculation (84Ivs0I), and 84 *vs.* 12 h post-inoculation (84Ivs12I). Data are from *Ayyappan et al. (2015)*.                               

the fact that many BAHD enzymes are involved in the biosynthesis of specialized metabolites, and considering the structure of PvHHHT products, these results support a potential role of this enzyme in abiotic and biotic stress responses.

We also looked for stress-responsive *cis*-elements in the promoter region of *Pvhhht*. Using the NewPlace database (*Higo et al., 1999*), we found elements related to abiotic stress, hormone, and pathogen responses (Fig. 5). Of note, there were two nodulation-related *cis*-elements (called OSE1 ROOTNODULE and OSE2 ROOTNODULE) found in the promoter, even though *Pvhhht* is not expressed in the nodules based on the Gene Expression Atlas data (*O'Rourke et al., 2014*) (Fig. 4A). A closer

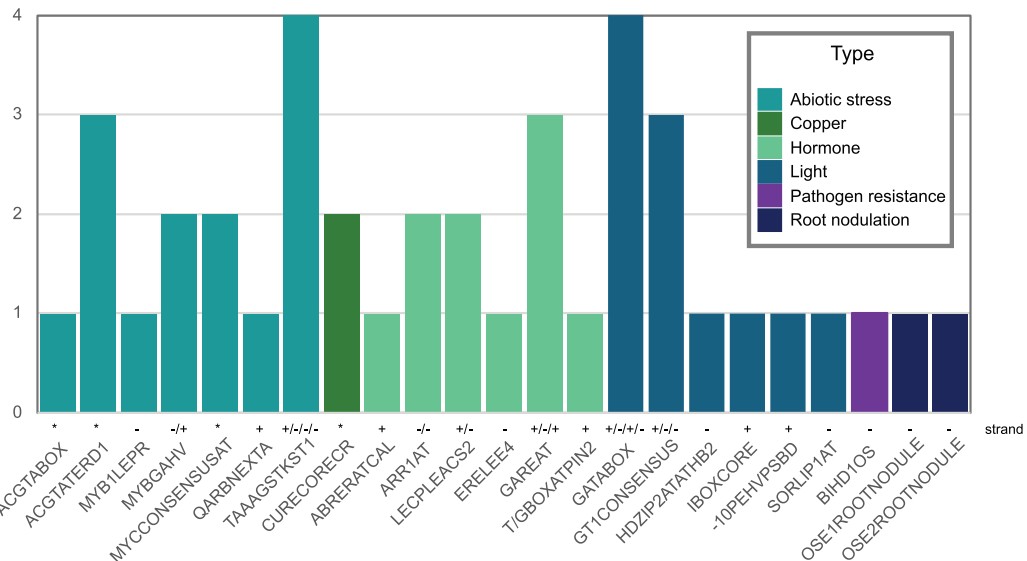

**Figure 5 Abiotic stress-, copper-, hormone-, light-, pathogenic-, and nodulation-responsive *cis*-elements present in the promoter sequence of *Pvhhht* (600 bp upstream of the transcriptional start).** Directionality of the *cis*-elements relative to the gene is noted above the name and called "strand." The *cis*-elements labeled with asterisks (*) are palindromes and therefore do not have a direction.

look at these motifs, specifically OSE2 ROOTNODULE, showed that it is oriented on the minus (−) strand approximately 133 base pairs upstream of the transcriptional start of *Pvhhht*. The orientation and sequence of this motif ('AAGAG') is an exact match to half of the nitrogen response element (NRE), which is the binding site of NIN-Like Proteins (NLPs) (*Jiang et al., 2021*). Furthermore, there is a possible match to the other half of the NRE 30 bp upstream of the 'AAGAG' motif, indicating that there may indeed be an NRE *cis*-element in the promoter of this gene that binds NLPs. While some NLPs have been shown to be important in the legume-rhizobium symbiosis, they are broadly responsive to the nitrogen status of the plant (*Jiang et al., 2021*; *Konishi & Yanagisawa, 2013*), which may be important for *Pvhhht* expression since it is upregulated under nitrogen deficiency conditions.

The role of PvHHHT products, hydroxycinnamoyl glucaric and saccharic esters, is still not clear. However, many of the characterized enzymes belonging to clade Vb are involved in the synthesis of phenolic metabolites known to be related to plant defense (*Roumani et al., 2021*; *Qiao et al., 2024*). In tomato, it has been suggested that caffeoyl glucaric acid esters have a role in pathogen response, as these inhibited the growth of tomato fruitworm (*Heliothis zea*) (*Elliger et al., 1981*). Another study showed that a caffeoyl glucaric acid derivative from the plant *Leontopodium alpinum* Cass. has pronounced antioxidative properties *in vitro* (*Schwaiger et al., 2005*). Together, these data (the HHHT products, the gene expression profile, and the *cis*-elements identified,) suggest PvHHHT participates in pathogen or abiotic stress responses, and provides a foundation for further work to address its role *in planta*.

## Determination of HHHT kinetic parameters

To analyze the enzymatic properties of HHHT, a histidine-tagged, codon-optimized version was expressed in *E. coli* and purified by immobilized metal affinity chromatography (IMAC) (Fig. 6). A large portion of the expressed protein was insoluble despite induction at 18 °C, which has been shown to enhance production of soluble protein for at least some hydroxycinnamoyl transferases (*Sullivan, 2009*). Nonetheless, soluble protein was apparent in the induced culture lysate and could be greatly enriched by IMAC. Migration of the induced and purified protein on SDS-PAGE was consistent with the predicted molecular mass of 52.6 kDa for the histidine-tagged HHHT. A major protein that copurified with HHHT (molecular weight of approximately 70 kDa) might be DnaK, which has been reported to copurify with histidine-tagged proteins on IMAC (*Rial & Ceccarelli, 2002*). The resulting HHHT preparation was used in kinetic analyses using *p*-coumaroyl-, caffeoyl-, and feruloyl-CoA donors and saccharic and mucic acid acceptors.

Initial reaction rates for HHHT were determined by measuring the release of free CoA spectrophotometrically using DTNB as previously described (*Sullivan, 2023*). To rule out any promiscuous reaction of DTNB with other components, we performed control reactions lacking either the donor or acceptor substrate, with the addition of enzyme and no substrate, and without added enzyme. We saw no significant net change in the absorbance at 412 nm for any of these controls.

Impact of pH on reaction rate was first determined in reactions with 20 µM *p*-coumaroyl-CoA donor and 2 mM saccharic acid as the acceptor at pH values between 6.5 and 8.0 (Table 3). Higher pH values were not tested as the CoA thioester becomes unstable at higher pH. Although the reaction rate was highest at pH 8.0, the rate at pH 7.5 was nearly as high (94%). Consequently, reactions to determine kinetic parameters for HHHT were carried out at pH 7.5 since the hydroxycinnamoyl-CoA donors are both more stable under this condition and have a lower extinction coefficient at 412 nm at this pH compared to pH 8.0, making rate measurements using DTNB simpler.

Results of the kinetic analysis are shown in Fig. 7 and Table 4. $K_M$ values for hydroxycinnamoyl donors were in the range from 5–20 µM, a range slightly higher than that reported for red clover HDT (*Sullivan & Knollenberg, 2021*), but lower than that reported for red clover HMT and HST (hydroxycinnamoyl-CoA: shikimate hydroxycinnamoyl transferase) (*Sullivan & Zarnowski, 2011*; *Sullivan, 2023*). $K_M$ values for *p*-coumaroyl- and feruloyl-CoA donors were similar for either acceptor substrate, but that of caffeoyl-CoA was markedly (>three-fold) higher with mucic acid as acceptor compared with saccharic acid. Values for $k_{cat}$ were two- to four-fold higher with saccharic acid as the acceptor, which also drove catalytic efficiency (as $k_{cat}/K_M$) to be higher for the saccharic acid acceptor compared to the mucic acid acceptor.

$K_M$ values for saccharic and mucic acid acceptors were in the 2.5 to 5 mM range for both acceptors and varied depending on which donor was being used. Values for $k_{cat}$ when acceptors were the variable substrate were nearly the same for each donor-acceptor combination as when donors were the variable substrate for each donor-acceptor

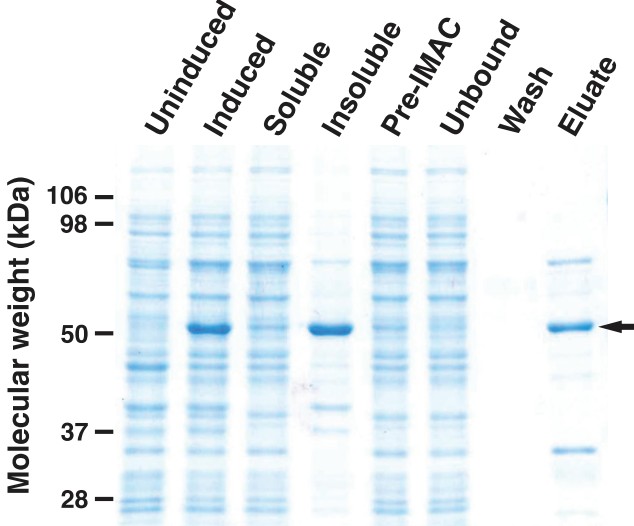

**Figure 6 SDS-PAGE gel (10%) showing expression of His-tagged HHHT in *E. coli* and purification by immobilized metal affinity chromatography (IMAC).** Lanes are uninduced culture, induced culture, soluble fraction, insoluble fraction, clarified soluble fraction applied to column (Pre-IMAC), unbound fraction, column wash, and eluate as indicated. All lanes were loaded with sample corresponding to 0.05 $OD_{600nm}$ of culture except eluate, which corresponds to 1.25 $OD_{600nm}$ of culture. The arrow on the right marks the migration of the His-tagged protein.

**Table 3 Effect of pH on reaction rate with 20 μM *p*-coumaroyl-CoA and 2 mM saccharic acid.**

| pH | Relative rate ± SEM |
| --- | --- |
| 6.5 | 0.46 ± 0.05 |
| 7.0 | 0.71 ± 0.02 |
| 7.5 | 0.94 ± 0.01 |
| 8.0 | 1.00 ± 0.01 |

combination. This is expected since this value represents turnover when substrates are at saturating levels. Because $K_M$ values for the acceptors are high relative to those of the donors, catalytic efficiencies (as $k_{cat}/K_M$) for these measurements are several hundred-fold lower than when donors were used as the variable substrate. Similar to measurements made with donors as the variable substrate, $k_{cat}/K_M$ were two- to four-fold lower for mucic acid than for saccharic acid. Although the higher catalytic efficiency observed for saccharic acid as an acceptor is consistent with the observation that hydroxycinnamoyl-saccharic acid esters predominate over mucic acid esters in bean leaves (*Sullivan, 2017*), acceptor concentration *in vivo*, currently unknown, would almost certainly also play a role in the relative accumulation of products *in vivo*, especially given the modest difference in magnitude of catalytic efficiencies.

Also, since the $K_M$ values for the acyl acceptors were relatively larger and considering that divalent metals commonly chelate carboxylates in active sites (*Larsen et al., 1996*), we tried adding $Mg^{+2}$ to the reaction, but this had no impact on the reaction rates.

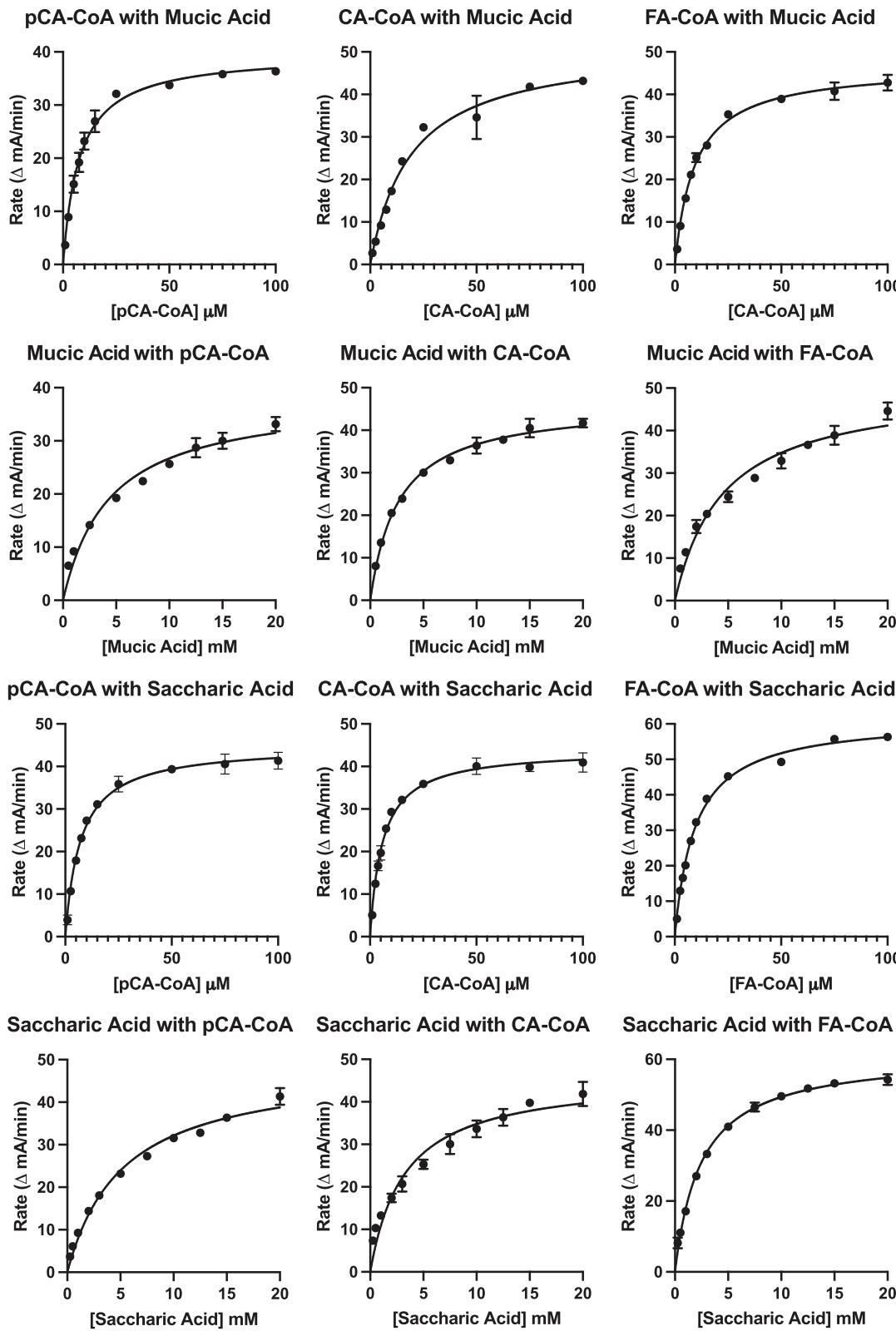

**Figure 7 Kinetic curves showing rate (as $\Delta mA_{412nm}$/min) vs. substrate concentration.** Donor and acceptor substrates are as indicated with pCA-CoA, CA-CoA, and FA-CoA indicating *p*-coumaroyl-, caffeoyl-, and feruloyl-CoA, respectively. $\Delta mA_{412nm}$/min values were converted to katal as described in *Sullivan (2023)* for calculation of $k_{cat}$.

**Table 4** Kinetic parameters for HHHT.

| Variable substrate[a] | Constant substrate[a] | $K_M$ (µM) | $k_{cat}$ (s$^{-1}$) | $k_{cat}/K_M$ (s$^{-1}$ mM$^{-1}$) |
|---|---|---|---|---|
| *p*CA-CoA | Saccharic acid (20 mM) | 7.2 ± 0.5 | 5.9 ± 0.1 | 820 ± 60 |
| CA-CoA | Saccharic acid (20 mM) | 5.8 ± 0.4 | 3.4 ± 0.1 | 590 ± 40 |
| FA-CoA | Saccharic acid (20 mM) | 9.6 ± 0.4 | 6.0 ± 0.1 | 630 ± 20 |
| *p*CA-CoA | Mucic acid (20 mM) | 7.7 ± 0.6 | 1.55 ± 0.03 | 200 ± 20 |
| CA-CoA | Mucic acid (20 mM) | 19 ± 2 | 1.3 ± 0.1 | 70 ± 9 |
| FA-CoA | Mucic acid (20 mM) | 9.4 ± 0.6 | 2.43 ± 0.04 | 260 ± 20 |
| Saccharic acid | *p*CA-CoA (100 µM) | 5,200 ± 500 | 6.4 ± 0.2 | 1.2 ± 0.1 |
| Saccharic acid | CA-CoA (100 µM) | 3,200 ± 500 | 3.6 ± 0.2 | 1.1 ± 0.2 |
| Saccharic acid | FA-CoA (100 µM) | 2,500 ± 100 | 6.0 ± 0.1 | 2.4 ± 0.1 |
| Mucic acid | *p*CA-CoA (100 µM) | 4,400 ± 700 | 1.5 ± 0.1 | 0.34 ± 0.05 |
| Mucic acid | CA-CoA (100 µM) | 2,600 ± 200 | 1.21 ± 0.03 | 0.46 ± 0.04 |
| Mucic acid | FA-CoA (100 µM) | 4,600 ± 600 | 2.6 ± 0.1 | 0.58 ± 0.09 |

**Note:**
[a] *p*CA-CoA, *p*-coumaroyl-CoA; CA-CoA, caffeoyl-CoA; FA-CoA, feruloyl-CoA.

## Protein structure model and molecular docking

To investigate the interactions between HHHT and the acyl group acceptor ligands (mucic and saccharic acid) in the active site, and gain insights about the higher catalytic efficiency of saccharic acid in comparison to mucic acid, we downloaded a predicted protein structure model of HHHT from AlphaFold and performed molecular docking of the ligands into the enzyme. The model had a high to very high confidence score for most of the residues. Molprobity results validated the structure. The predicted HHHT structure has two chloramphenicol acetyltransferase-like domains (residues 1 to 207 and residues 214 to 448) and consists of 14 beta-sheets and 19 alpha-helices (Fig. 8). The two-domain 3-D structure is similar to that of other BAHD acyltransferases elucidated using X-ray crystallography (*Lallemand et al., 2012*; *Ma et al., 2005*; *Walker et al., 2013*). The active site motif HXXXD is in a region predicted with a very high-confidence score by AlphaFold. Nevertheless, some regions close to the active site, predicted with a high confidence score, carry some amino acids that could participate in conformational changes or interactions with other components, which cannot be predicted by AlphaFold. For example, the region from amino acids 345–351 contain a solvent exposed arginine that could form salt bridges. The region from residues 351–375 line the active site and carry a solvent exposed cysteine (Cys-359), which could be involved in the formation of different conformations or stabilizing quaternary structure *via* disulfide bridges. There are also other residues such as Glu-354, Asp-358, Arg-361, Arg-362, that could form salt bridges, and Met-365 that could form disulfide bonds. Therefore, this region is likely flexible.

After performing the molecular docking of the mucic acid and saccharic acid structures into the protein model, nine conformations were obtained for each ligand (Table 5). The lowest binding energies obtained for saccharic and mucic acid conformations were very similar, which suggests there is not a markedly higher affinity for one substrate over the

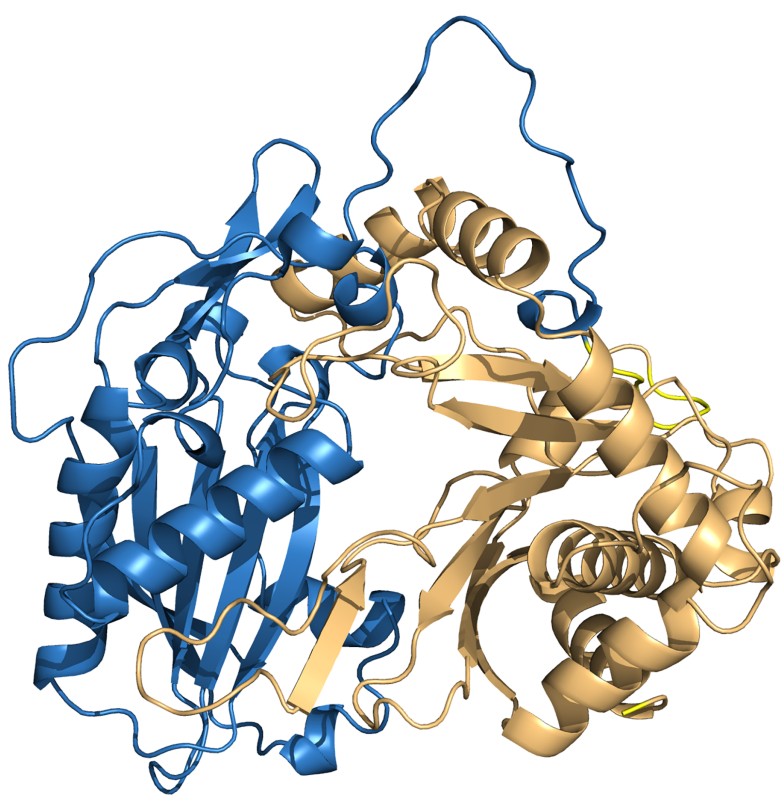

**Figure 8** **PvHHHT structure predicted by alphafold.** The two chloramphenicol acetyltransferase-like domains are colored in blue and light orange.

**Table 5** **Affinity (kcal/mol) for each of the nine conformations resulting from the molecular docking of mucic acid or saccharic acid into the HHHT protein using AutoDock Vina.**

| Mode | Mucic acid | Saccharic acid |
|---|---|---|
| 1 | −5.29 | −5.621 |
| 2 | −5.148 | −5.556 |
| 3 | −5.093 | −5.528 |
| 4 | −5.063 | −5.498 |
| 6 | −4.992 | −5.368 |
| 7 | −4.978 | −5.367 |
| 8 | −4.898 | −5.353 |
| 9 | −4.876 | −5.309 |

other. This is also consistent with the measured $K_M$ values for the acceptors, which were very similar for saccharic and mucic acid. The lowest energy conformations (number 1 in the table) were visualized to analyze the HHHT active site (Fig. 9). The protein residues His 150 and Gly 155, which are part of the HXXXD motif, form interactions with both mucic and saccharic acid. It has been shown for many BAHD acyltransferases that the histidine in this motif is essential for activity. In multiple mutagenesis experiments,

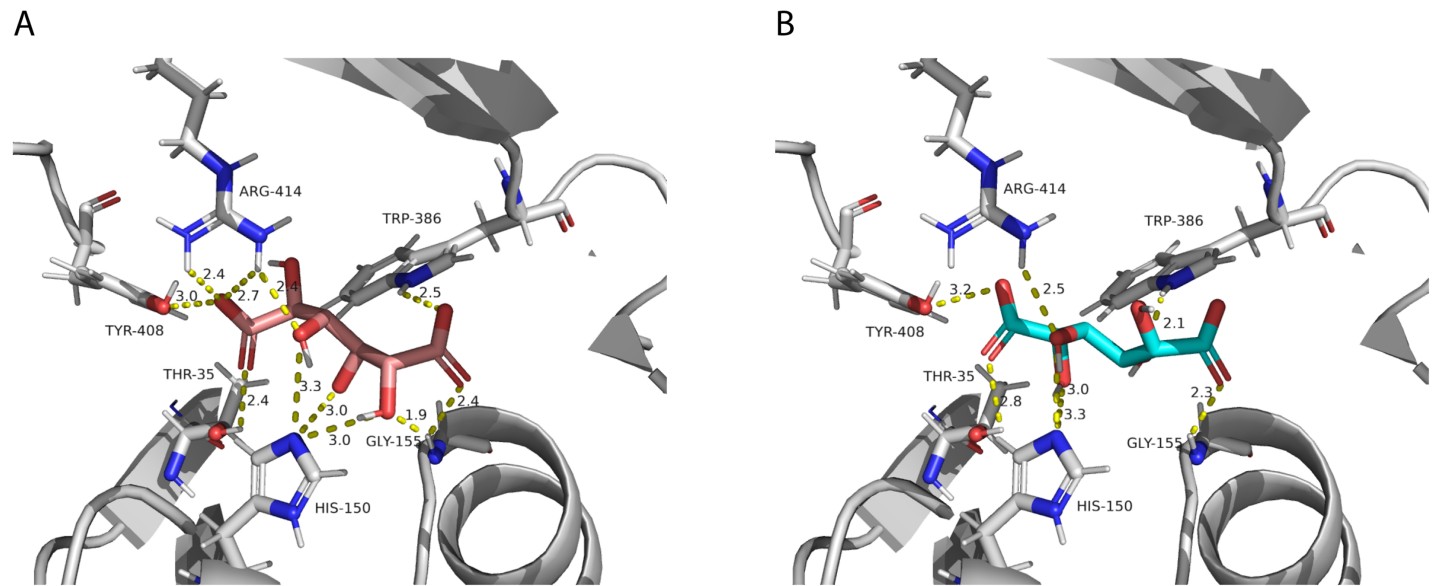

**Figure 9  PvHHHT polar interactions with acyl acceptor ligands in the active site.** (A) Saccharic acid. (B) Mucic acid.

replacement of this histidine with alanine abolishes enzyme activity (*Lallemand et al., 2012*; *Walker et al., 2016*; *Bayer, Ma & Stöckigt, 2004*). It has been proposed that this catalytic histidine serves as a general base, abstracting a proton from the acyl acceptor substrate (*Levsh et al., 2016*; *Ma et al., 2005*). For sorghum HCT (*Walker et al., 2016*), it was proposed that the imidazole atom from His 152 is properly oriented with the aid of Thr 36 to abstract a proton from shikimic acid. This enables a nucleophilic attack of the shikimate hydroxyl on the thioester carbonyl of hydroxycinnamoyl-CoA. Here, with the docking results, we also found Thr 35 in the active site, next to His 150 that could be aiding the proton abstraction from a hydroxyl group of mucic or saccharic acid. It was also found for SbHCT that a Trp in position 384 and an Arg in position 371 are important for catalytic activity and are likely involved in the binding of the shikimate molecule. We also found Trp in position 386 and Arg in position 414 of HHHT interacting with the substrates in our docking analysis. Another amino acid we found interacting with mucic and saccharic acid was Tyr 408.

We also investigated potential binding sites for the hydroxycinnamoyl ligands. Since these molecules are large, which represents a challenge for accurate computational docking (*Devaurs et al., 2019*), we did not obtain satisfactory results using the whole ligand. Therefore, we used the N-acetyl-S-hydroxycinnamoyl-cysteamine portion for the analysis. We obtained nine conformations for each acyl donor ligand (Table 6). The lowest binding energies for *p*-coumaroyl, caffeoyl and feruloyl ligands were all similar, which is consistent with the similar donor substrate $K_M$ values. In these conformations, they were all placed in the same region as mucic and saccharic acid (Fig. 10), which supports the identified binding site. The acyl-donors formed interactions with Thr 37, Gly 155 and Arg 414.

**Table 6 Affinity (kcal/mol) for each of the nine conformations resulting from the molecular docking of N-acetyl-S-hydroxycinnamoyl-cysteamine into the HHHT protein using AutoDock Vina.**

| Mode | Caffeoyl | p-Coumaroyl | Feruloyl |
|------|----------|-------------|----------|
| 1 | −6.953 | −7.010 | −6.919 |
| 2 | −6.946 | −6.860 | −6.845 |
| 3 | −6.824 | −6.740 | −6.778 |
| 4 | −6.608 | −6.588 | −6.652 |
| 6 | −6.535 | −6.574 | −6.583 |
| 7 | −6.464 | −6.484 | −6.569 |
| 8 | −6.297 | −6.237 | −6.441 |
| 9 | −6.159 | −6.167 | −6.355 |

All the putative residues identified through docking as interacting with ligands are conserved in red clover HMT and HDT as well, with exception of Thr 35 which is replaced by a valine in HDT (Fig. 3), suggesting they are important for transferase activity.

In our analysis, we were not able to determine to which hydroxyl group of mucic or saccharic acid the hydroxycinnamate is transferred. Nonetheless, for the imidazole nitrogen of His 150 to abstract a proton from the hydroxyl group, the distance between the atoms has to be less than 4 Å (*Harris & Mildvan, 1999*). From the docking (Fig. 9), for mucic acid, hydroxyl groups in C4 and C5 are positioned 3.0 and 3.3 Å, respectively, from the imidazole nitrogen, which would allow a proton transfer. For saccharic acid, hydroxyl groups in C2, C3, and C4 are 3.0, 3.0. and 3.3Å, respectively, from the imidazole nitrogen, and could be involved in the proton transfer reaction. In previous work (*Sullivan, 2017*), we analyzed PvHHHT *in vitro* reaction products using LC-MS and observed multiple peaks. We detected the formation of three to four different products for the *in vitro* reaction of p-coumaroyl-, caffeoyl- or feruloyl-CoA with saccharic acid. On the other hand, we detected only one product for the reaction of p-coumaroyl-CoA or caffeoyl-CoA with mucic acid, and two products for FA with mucic acid. We considered this to be likely the result of non-enzymatic intramolecular transesterification reactions between the hydroxycinnamoyl-ester and the other hydroxyl groups of the molecule. However, the larger diversity of products identified for the reactions with saccharic acid *vs*. mucic acid, together with the docking results showing proximity of multiple hydroxyl groups to the imidazole N of His 150, suggest that besides transesterification reactions it may be that HHHT can acylate different sites of the tetrahydroxyhexanedioic acids molecules. BAHD transferases catalyzing multisite acylation reactions have been previously reported. *Arabidopsis thaliana* AtSHT and AtSDT, also belonging to clade Vb (Fig. 2), substitute multiple N positions of spermidine (*Wang et al., 2021*). Other spermidine hydroxycinnamoyl transferases involved in the synthesis of tri-substituted spermidine or tetra-substituted spermine were identified in *Malus domestica* (MdSHT) and *Cichorium intybus* (CiSHT) (*Elejalde-Palmett et al., 2015*; *Delporte et al., 2018*). Also, a saponine acetyltransferase from *Astragalus membranaceus*, AmAT7-3, acetylates C3′-*O* and C4′-*O*

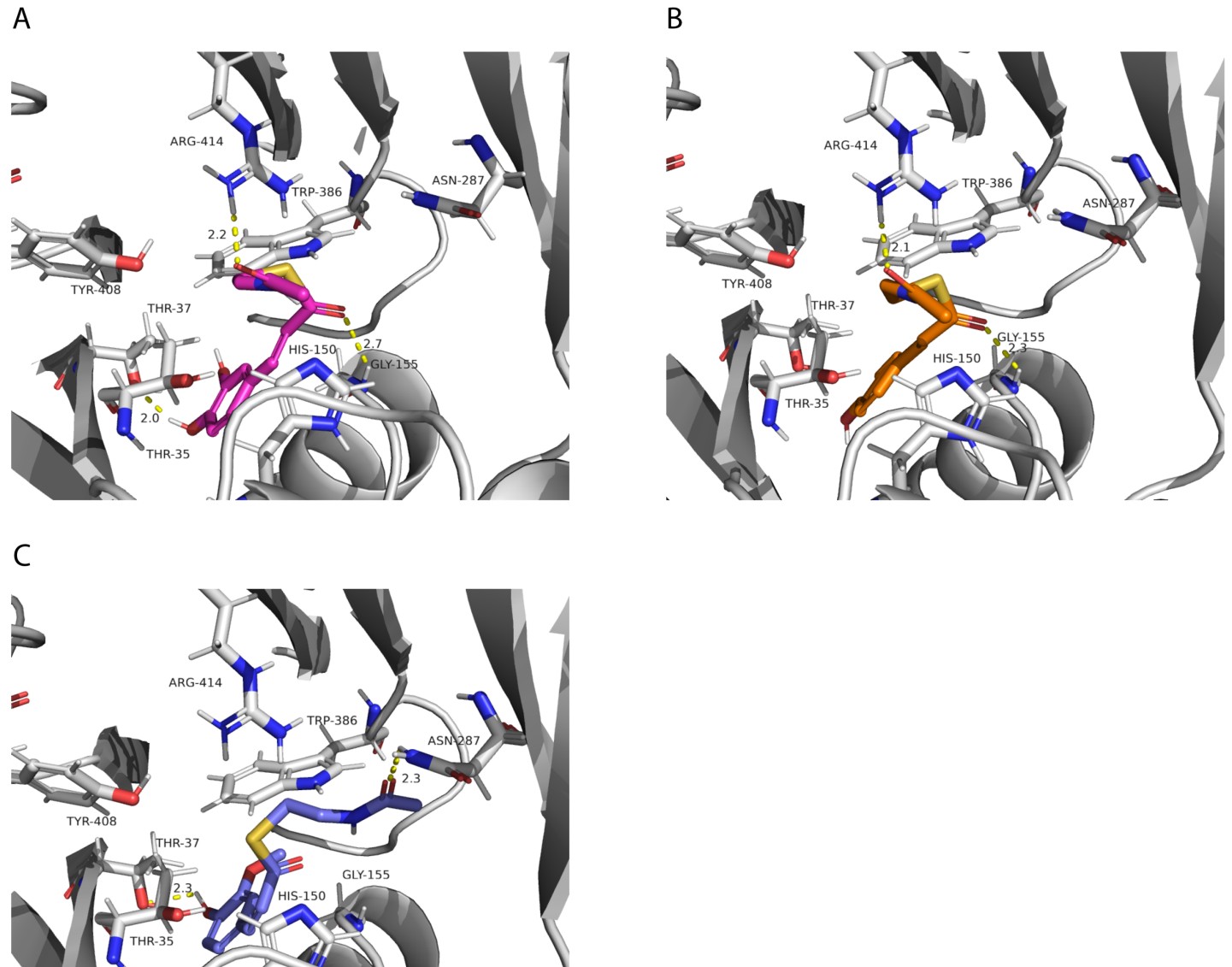

**Figure 10 PvHHHT polar interactions with acyl donor ligands in the active site.** (A) Caffeoyl. (B) *p*-Coumaroyl. (C) Feruloyl.

of xylose of the compound astragaloside IV (*Wang et al., 2023*). Therefore, PvHHHT could also be a multi-site transferase.

### Site-directed mutagenesis and enzymatic analysis

To test whether the interacting residues identified through the structure and docking analysis are important for substrate binding or catalysis, we used site-directed mutagenesis to replace each of the residues Thr 35, His 150, Trp 386, and Arg 414 with alanine and evaluated transferase activity for the reaction with 100 μM *p*-coumaroyl-CoA and 20 mM saccharic acid. All mutants had very little activity, ranging from 2% to 8% of that of the

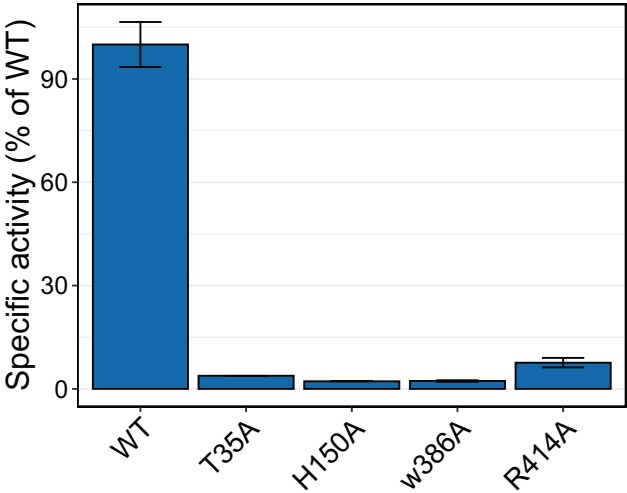

**Figure 11 Relative specific activities of HHHT mutants compared to wild type.** Activities were measured with 100 μM $p$-coumaroyl-coA and 20 mM saccharic acid. One hundred percent corresponds to 98.6 Katal/mg.

native wild-type protein (Fig. 11). These results support the docking analysis and show that these residues are important for catalysis.

The impact of the H150A mutation supports the important catalytic role of this residue, which is part of the HXXXD active site motif. In their work with SbHCT, *Walker et al. (2016)* also found very little activity for a T36A and a R371A mutant. Also, R386 in the active site of *Arabidopsis thaliana* HCT was shown to be important for substrate specificity (*Levsh et al., 2016*).

## CONCLUSIONS

Like other clade Vb BAHD transferases, expression pattern and promoter analysis suggest a role of HHHT in defensive responses. The highest level of expression appears to be in leaves of plants inoculated with an ineffective strain of nitrogen-fixing rhizobia (Fig. 4A). HHHT was also upregulated in response to inoculation with a fungal pathogen (Fig. 4B). Further, numerous potential stress-responsive promoter elements are present in the upstream region of the gene (Fig. 5). Kinetic parameters for both donor CoA substrates (μM $K_M$) and acceptor substrates (mM $K_M$) are in the range of those reported for other hydroxycinnamoyl transferases that have been biochemically characterized (Table 4). The enzyme appears to be more efficient in using saccharic acid compared to mucic acid, even though they had similar binding poses when docked into the active site and their $K_M$ did not vary substantially. This suggests that factors other than acceptor substrate affinity contribute to the difference in efficiency between the two substrates. For example, mucic acid may be less favorably oriented in the active site for ester formation relative to saccharic acid. Furthermore, proximity of multiple hydroxyl groups of saccharic acid to the imidazole N of the active site His may explain the formation of multiple products previously attributed to intramolecular transesterifications. The structure and active site data here, along with kinetic parameters of the enzyme should contribute to the growing

body of such data that might ultimately provide a better understanding of the relationship of the 3-D structure of BAHD transferases to their substrate specificity, which should allow better prediction of enzyme function from primary amino acid sequence. These data could further provide the basis for rational design of BAHD enzymes to produce desired and novel products.

## ACKNOWLEDGEMENTS

We wish to thank Joe Jez for useful discussions on structure modeling and active site docking. We also thank Laurie Reinhardt for helpful comments on the manuscript. All opinions expressed in this article are the authors' and do not necessarily reflect the policies and views of USDA, DOE, or ORAU/ORISE. Mention of trade names or commercial products in this article is solely for the purpose of providing specific information and does not imply recommendation or endorsement by the U.S. Department of Agriculture.

### Funding

This work was supported by ARS Projects 5090-21000-071-00D and 5090-21500-001-00D. This research was supported by the appointment of Amanda Fanelli to the Agricultural Research Service (ARS) Research Participation Program administered by the Oak Ridge Institute for Science and Education (ORISE) through an interagency agreement between the U.S. Department of Energy (DOE) and the U.S. Department of Agriculture (USDA). ORISE is managed by ORAU under DOE contract number DE-SC0014664. There was no additional external funding received for this study. The funders had no role in study design, data collection and analysis, decision to publish, or preparation of the manuscript.

### Grant Disclosures

The following grant information was disclosed by the authors:
ARS Projects 5090-21000-071-00D and 5090-21500-001-00D.
Agricultural Research Service (ARS).
Oak Ridge Institute for Science and Education (ORISE).
U.S. Department of Energy (DOE).
U.S. Department of Agriculture (USDA).
ORAU: DE-SC0014664.

### Competing Interests

The authors declare that they have no competing interests.

### Author Contributions

- Amanda Fanelli conceived and designed the experiments, performed the experiments, analyzed the data, prepared figures and/or tables, authored or reviewed drafts of the article, and approved the final draft.
- Christina Stonoha-Arther analyzed the data, authored or reviewed drafts of the article, and approved the final draft.

• Michael L. Sullivan conceived and designed the experiments, performed the experiments, analyzed the data, prepared figures and/or tables, authored or reviewed drafts of the article, and approved the final draft.

## Data Availability

The biochemically characterized BAHD belonging to clade Vb are available at NCBI: 4KE4_A, NP_001408294.1, XP_027125824, UHJ19789.2, LEKV01000029.1, XP_024963804.1, KT222891, KT222892–KT222895, MG457243, MG457244, XM_022184559.1, XM_022139600.1, ABI48360, NP_001306184.1, NP_001411209.1, NP_001403641.1, NP_001408293.1, NP_179497.1, BB926056.1, CAB06430, BAC78633, ABO52899, CAD47830, XP_045828072.1, ACI16630.1, NP_199704, CAE46932, CAE46933, ABK79689.1, ABK79690.1, CAK55166.

The Python script used to manipulate the FASTA files for phylogenetic analysis, raw output of the phylogenetic analysis of the BAHD family, R scripts and the dataset used for gene expression profile analyses and the raw data used in kinetic analyses are available in Supplemental Files.

## Supplemental Information

Supplemental information for this article can be found online at http://dx.doi.org/10.7717/peerj.19037#supplemental-information.

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
