# Peer review of "Hydroxycinnamoyl-coenzyme A: tetrahydroxyhexanedioate hydroxycinnamoyl transferase (HHHT) from Phaseolus vulgaris L.: phylogeny, expression pattern, kinetic parameters, and active site analysis"

_PeerJ, doi:10.7717/peerj.19037_

## Round 0.1 · original submission · Major Revisions

Please take note of the detailed comments from the reviewers and submit a revised version along with a rebuttal letter addressing the peer's comments.

·

Basic reporting

In the manuscript, the authors describe the expression pattern, kinetic parameters, structure analysis, and active site interactions to better understand the HHHT from Phaseolus vulgaris L. The manuscript has a good and clear structure with sufficient background to strengthen the results.
While the manuscript is of high quality, some areas could be enhanced without compromising its publication.
In line 40, the authors could expand the information to describe better how the structure is related to substrate specificity. Some of this is described in line 50.

Experimental design

The authors describe all methods according to the objective and results, with sufficient details.
I suggest some additions:
Incorporate criteria for visual inspection (line 120).
In line 123, authors should specify which set of transferases were used.

Validity of the findings

The results are consistent and well-stated. I suggest some additions in the discussion.
The redaction in line 232 needs to be more precise; authors need to specify some roles. This case is also observed in lines 255 for the subclade and roles and 259 for the products.
Paying attention to unsoluble protein is unnecessary if it is not used in work (line 312).
The structure of lines 341-344 needs to be clarified; I suggest separating it into sentences.

Additional comments

No additional comments

Reviewer 2 ·

Basic reporting

In this manuscript untitled "Hydroxycinnamoyl-coenzyme A:tetrahydroxyhexanedioate hydroxycinnamoyl transfrease (HHHT) from Phaseolus vulgaris L.: phylogeny, expression pattern, kinetic parameters, and active site analysis", authors present the study of a previously characterized enzyme that is involved in the acylation of mucic and glucaric acids in bean.
I think this article suffers from the lack of novelty and in addition, most of the conclusions are very speculative and do not rely on experimental data.
English is clear and unambiguous and the article is well written.

Experimental design

No comment

Validity of the findings

The activity of this enzyme was already assessed both in vitro and in vivo in a previous paper published in the journal Planta (Sullivan, 2017) and the main conclusion of the present paper i.e. catalytic activity of HHHT is already well documented in this former paper.

For the phylogenetic analysis (Figure 1 and experimental parts), authors used 69 previously biochemically chararacterized BAHD proteins which were identified before 2011 (line 118). I think authors should update their analysis with the numerous enzyme sequences that have subsequently been identified. This will undoubtedly improve their analysis and subsequent analysis. Furthermore, they used 3 leguminous, Arabidopsis and 1 Monocot for their analysis and the construction of the phylogenetic tree. They conclude that enzymes from the leguminous form a distinct cluster (distinct from SHT of Arabidopsis). They should add more sequences from different plant family to enforce their findings. For example, where is the recently identified EpHTT from Equinaceae purpurae (Fu et al. 2021)? There are too much leguminous sequences to correctly support the tree.

The transcriptomic analysis relies on two previously published datasets. One is related to nitrogen nutrition and plant development and the other to pathogen infection (Figure 4). Authors should generate their own set of data using qPCR to confirm these RNAseq data. They observed correlation between the gene expression and nitrogen deficiency as well as pathogen attack. From that, they deduced that HHHT is involved in biotic/abiotic stress responses. This hypothesis should be support by experimental data. Otherwise, this remains highly speculative. I think the same criticism could be made when authors described detection of cis-elements in the promoter region of PvHHHT. Because of the presence of predicted cis-elements related to abiotic or biotic stresses, authors consider that this is enough to predict a role of this gene in such conditions. Again, more experiments are needed.

Kinetic parameters were determined spectrophotometrically. Giving that the colorimetric assay is not very specific, LC-MS data (or any other analytical technique) should be provided to assess the presence and the identity of the products of the reaction. Authors mentioned that their analyses were carried out in duplicates (line 203). Nevertheless, in Figure 7, there are means and SD. Were the analyses performed at least in triplicate instead of duplicate?

Table 4 summarizes the kinetic parameters measured with the different substrates. What can be conclude from this analysis? Did the authors made competition assays (equimolar concentrations of both acyl acceptors) to see what is the best substrate (mucic or saccharic acid)?

From the molecular docking, some amino acids are important for catalysis or for the stabilization of the substrates. This should be supported by experimental data. Authors should think about a mutagenesis approach to demonstrate their findings.

Minor issues :
Line 83 : "coumaryl-CoA" should be replaced by coumaroyl-CoA

·

Basic reporting

The manuscript titled “Hydroxycinnamoyl-coenzyme A:tetrahydroxyhexanedioate hydroxycinnamoyl transferase (HHHT) from Phaseolus vulgaris L.: phylogeny, expression pattern, kinetic parameters, and active site analysis” describes the thorough characterization of a representative member of a BAHD acyl-CoA transferase family. The manuscript is mostly well written.

Experimental design

The authors previously confirmed the activity of HHHT. Here they characterize the enzyme kinetics and substrate specificity in detail, a meaningful endeavor. However, the expression profiling and promoter analysis is highly speculative and disconnected from the main theme of kinetics and structural analysis. There are some concerns with both the kinetics analysis and modeling that can be addressed. Some information is needed to support replication. Those concerns are detailed in the additional comments.

Validity of the findings

The experiments are mostly straightforward, however some of the caveats associated with the use of DTNB have not been addressed, which may lead to different conclusions. A few control reactions are suggested in the additional comments.

Additional comments

Some concerns/suggestions:
1) DTNB reacts with cysteines that are exposed on the surface of proteins. Was any attempt made to control for the generation of disulfides or mixed TNB-enzyme disulfides? HPLC analysis of a couple control reactions of one donor-acceptor pair at 10-20% completion under reducing conditions should be compared with the DTNB assay to determine if the rates are altered by TNB-enzyme adducts. Since reductants are never used in the purification, one could assume that some of the thiols have already formed symmetrical disulfides, disulfides with DnaK or with glutathione. Analysis on SDS-PAGE in the absence of reductant could reveal the proportion that is oxidized to other forms. Any of these oxidized forms could alter the substrate specificity through allosteric mechanisms.
2) The enzyme is significantly contaminated with DnaK which commonly co-purifies with proteins expressed in E. coli. The authors should acknowledge this in the results/discussion. https://doi.org/10.1016/S1046-5928(02)00024-4
3) The low affinity for the acceptor substrates suggests something is missing. Have the authors tried divalent metals such as magnesium which commonly chelate carboxylates in enzyme active sites?
4) The docking was previously reported in a Methods in Enzymology chapter. Figure 1 contains the same structures of the acceptor molecules with the same exact errors in that both have carboxylic acids where both oxygens are protonated.
5) The regiospecificity is not determined. The location of the transferred cinnamate would be extremely useful for understanding the structure-function relationship and any modeling. Statements as to why this was not done in the results/discussion would be appropriate.
6) The regions of the AlphaFold model that are close to the active site with “High” confidence should be examined carefully. These regions are likely flexible and adopt multiple conformations. For example, residues 351-375 form regions that line the active site cavity and carry a solvent exposed cysteine. This is completely passed over in the manuscript as a potential place that could confound analysis.
7) Typically modelling is accompanied by mutagenesis to determine if the predicted roles are true. For example, the statement on lines 367-370, suggesting certain residues are important for substrate binding could be tested by mutagenesis to validate the docking.
8) Cinnamoyl-CoA is not used in the docking or modeled in any way. Even modeling a portion of cinnamoyl-CoA (as cinnamoyl N-acetyl cysteamine) could reveal if the binding site for the donors is an artifact of an active site that does not have the high affinity substrate bound first.
Line 27, “appears to have involved both divergent” changing involved to “arisen from” might be clearer.
Lines 37-40, we encourage the authors to not make conclusions from the docking, furthermore, this sentence is not helpful for understanding how the enzyme works.
Line 47, BAHD acyltransferase
Line 52, Only 7 clades are mentioned
Line 114, E-value. Higher than 10^-5. Higher exponents in this case are lower/small values. Do the authors mean greater significance?
Line 161, which pET-28 vector? pET-28a(+)?
Line 167, antibiotics?
Line 193, real time, continuous assay
Line 196, please state DTNB concentration
Line 197, “As needed” is ambiguous. One is acid form other is potassium salt already. By adding an equivalent of KOH for the potassium salt and two equivalents for the acid form?
Line 210, “PDB file AF-V7BKA6-F1-v4”?
Lines 212-213. The molecules were not “designed”, rendered or encoded?
Lines 213-214. What are the predicted or known pKa’s for the carboxylates? It’s likely that both are below 7.4. Succinate or malate in comparison have pKa’s below 6 for all carboxylates. In figure 9 both appear to be completely deprotonated, but it should be stated clearly. As written it reads as hydrogens were added, rather than removed.
Line 32, “exhaustiveness was set to 32”. Either here or in the results/discussion please state that this means bonds were allowed to rotate.
Line 348, since in vivo concentrations of substrates dictate relative turnover for substrates with similar kinetics, a statement that substrate concentration in vivo is unknown would be useful here.
Line 355, alpha-fold, AlphaFold
Line 368, why are the amino acid residues presented in a random order instead of sequentially?
Line 369, how does an enzyme “stabilize” a substrate? Do the authors mean orient? Enzymes stabilize the transition state not substrates.
Lines 377-378, this sentence could be better written. “…attack of the shikimate hydroxyl on the thioester carbonyl of…”
Lines 382-384, this sentence is a repeat of line 368.
Lines 384-385, this sentence is obvious, most residues near substrates play a role in catalysis. The important question is what roles do they play? Is there something new that was learned from the modeling? If so, it should be summarized here. It is ok if the modeling was unhelpful as there is agreement between the binding poses not differing between the acceptors and the Km’s also not differing significantly.
Lines 396-397, the Km/kcat values have errors that are as large as the values, how is this a significant difference? Especially considering the differences in Km’s are only around two-fold and the differences in kcat are only 3 fold. These differences are barely significant.
Figure 3. Please annotate potential substrate binding residues to allow comparison with homologs.
Figure 9. The images are not the same size, and appear to be in slightly different orientations. It gives the impression that the protein is moving.
Table 1. Are three significant figures for protein concentration needed?
Tables 3 and 4. Too many (or few) digits for the values and errors. The errors should be a single digit and the value should be reported to the single digit of the error. Changing the Km to mM will make it easier to report. The errors on kcat/Km appear to be too large.

---

## Round 0.2 · Minor Revisions

A reviewer noted a few minor aspects that need your attention. Please address them at your convenience.

·

Basic reporting

The authors attended to the review suggestions in the manuscript and incorporated a clear structure and additional background to strengthen the results.

Experimental design

The authors incorporate additional explanations for the visual inspection criteria and added the set of transferases used as an outgroup as suggested.

Validity of the findings

The authors included precise specifications in the manuscript structure, augmented the information in the study, and rearranged the sentences to improve clarity.

Additional comments

No additional comments

Reviewer 2 ·

Basic reporting

I think authors made major improvements in accordance with my comments. For instance, the mutagenesis approach greatly support their structural analysis. I am still not convince that the transcriptomic and the promoter analysis are needed. They are not supported by experimental data. They are too much speculative and the interpretation based on these data are speculative as well. Otherwise, the phylogenetic and kinetic analyses are now well presented.

Experimental design

No comment

Validity of the findings

No comment

Additional comments

Line 84-85: "Such as Alfalfa" is written twice
Line 262-263: EPHMT transfers hydroxycinnamates to tartric acid, forming caftaric acid
Line 262: Equinacea
Line 297-298: EpHMT and TpHMT do not catalyze the same reaction so this example is not appropriate to highlight the discrepancy between sequence homology and enzymatic activity
Line 488 : formation of tetrasubstituted spermidine is not possible. CiSHT catalyzes the tetrasubstitution of spermine.

·

Basic reporting

The authors have addressed my concerns.

Experimental design

The authors have addressed my concerns.

Validity of the findings

The authors have addressed my concerns.

---

## Round 0.3 · Minor Revisions

The updated version incorporates all reviewer feedback. I have noted several typos and included some recommendations for your consideration. Please refer to the attached PDF with these suggestions, and when you're ready, submit the final revised version.

---

## Round 0.4 · accepted · Accept

Thanks for considering the final details; your manuscript is now accepted in PeerJ!